# Sex and Sexuality in Autism Spectrum Disorders: A Scoping Review on a Neglected but Fundamental Issue

**DOI:** 10.3390/brainsci12111427

**Published:** 2022-10-24

**Authors:** Maria Grazia Maggio, Patrizia Calatozzo, Antonio Cerasa, Giovanni Pioggia, Angelo Quartarone, Rocco Salvatore Calabrò

**Affiliations:** 1Department of Biomedical and Biotechnological Science, University of Catania, 95123 Catania, Italy; 2Studio di Psicoterapia Relazionale e Riabilitazione Cognitiva, 98124 Messina, Italy; 3Institute for Biomedical Research and Innovation, National Research Council of Italy (IRIB-CNR), 98164 Messina, Italy; 4Sant’Anna Institute, 88900 Crotone, Italy; 5Pharmacotechnology Documention and Transfer Unit, Preclinical and Traslation Pharmacology, Department of Pharmacy, Health Science and Nutrition, University of Calabria, 87036 Calabria, Italy; 6IRCCS Centro Neurolesi “Bonino Pulejo”, 98121 Messina, Italy

**Keywords:** autism spectrum disorder, gender dysphoria, sexual awareness, sexual behavior, sexual identity, sexual orientation

## Abstract

ASD consists of a set of permanent neurodevelopmental conditions, which are studded with social and communication differences, limited interests, and repetitive behaviors. Individuals with ASD have difficulty reading eye gestures and expressions, and may also have stereotyped or repetitive language, excessive adherence to routines, fixed interests, and rigid thinking. However, sexuality in adolescents and young adults with ASD is still a poorly studied and neglected issue. This review aims to evaluate sexual function and behavior in individuals with ASD to foster a greater understanding of this important, although often overlooked, issue. This review was conducted by searching peer-reviewed articles published between 01 June 2000 and 31 May 2022 using the following databases: PubMed, Embase, Cochrane Database, and Web of Science. A comprehensive search was conducted using the terms: “Autism” OR “ASD” AND “Sexuality” OR “Romantic relation” OR “sexual behavior” AND/OR “sexual awareness”. After an accurate revision of 214 full manuscripts, 11 articles satisfied the inclusion/exclusion criteria. This review found that, although individuals with ASD may have sexual functioning, their sexuality is characterized by higher prevalence rates of gender dysphoria and inappropriate sexual behavior. Furthermore, sexual awareness is reduced in this patient population, and the prevalence of other variants of sexual orientation (i.e., homosexuality, asexuality, bisexuality, etc.) is higher in adolescents with ASD than in non-autistic peers. Sexual health and education should be included in the care path of patients with ASD in order to improve their quality of life and avoid/reduce inappropriate and risky behaviors.

## 1. Introduction

Autism spectrum disorder (ASD) is a developmental disability characterized by alterations in social interaction, verbal/non-verbal communication, and behavior [1]. The *Diagnostic and Statistical Manual of Mental Disorders—5th Edition* (DSM-5) defines ASD as the occurrence of persistent impairments in social interaction and the presence of limited and repetitive patterns of behavior, interests, or activities [2]. Individuals with ASD have difficulty in social communication, such as reading eye gestures and expressions [3,4] and may also have stereotyped or repetitive language, excessive adherence to routines, fixed interests, and rigid thinking. Social cognition, which concerns the detection, processing, and use of social information to regulate interpersonal functioning and effective social behavior, is particularly affected in ASD [2,3,4]. ASD individuals may have different cognitive profiles, and, in some cases, they may present atypical sensory perception and difficulties in processing information and motor skills. However, ASD is heterogeneous and exhibits a wide spectrum of intellectual abilities [2].

ASD prevalence estimates have increased in recent years, and approximately 1–2% of the population is diagnosed with ASD [5]. ASD is widespread and an estimated 1 in 44 children has been identified with this diagnosis according to CDC Autism and Developmental Disabilities Monitoring Network estimates [5]. ASD is more common among boys than among girls (3/4:1) [5], and the syndrome occurs early in life with symptoms appearing within the first 2 years. Young patients may have little ability to interact with others and engage in romantic relationships [6]; therefore, ASD individuals may experience difficulties in sexual interactions as they grow up [7,8,9].

To deal with this problem, it is essential to consider data on normative sexuality in adolescents. Ongoing studies on the topic have mainly focused on sexual behaviors, while other components, such as sexual desire, function, arousal, and experience, are poorly investigated, and they have been mostly considered only in the adult population [10,11,12,13,14]. Studies on the development of sexuality have found that the onset of puberty in adolescents is earlier than decades ago. In addition, the age for sexual debut has also advanced, both for masturbatory acts and in couples, which starts from 12 years [10,13]. Additionally, studies report that most teens define themselves as heterosexual, although uncertainty about sexual orientation is on the rise [10,12], and new policies are needed to improve the mental health of gender and sexual minorities [14]. On the other hand, the studies on the topic of ASD are still poor and very confusing.

Decades ago, people with ASD were thought to be “asexual” [15]. However, it is known that they experience an intense desire for love, and romantic and/or sexual relationships, especially in highly functioning young people [16,17,18]. Some studies have shown that the interest shown/expressed by individuals with ASD can be similar to that of N-ASD individuals, and the substantial difference lies in the completeness of the relational experience [19,20]. In summary, the relationship difficulties resulting from ASD do not allow individuals to relate adequately to others, leading to unwanted sexual contact, social isolation, and mental health disorders [21,22]. Indeed, Stokes et al. have shown that people with ASD fail to learn various aspects of sexual functioning, as occurs in peers, preventing them from obtaining adequate sex education due to fewer opportunities for social contact [18]. Indeed, information on sexual well-being and behavior is often provided by parents or educators [23]. In recent years, the use of technologies and the Internet to obtain more information on sexuality by people with ASD has also become increasingly widespread [24,25,26]. However, this can carry the risk of inadequate and unclear ideas and norms, which can incentivize inappropriate behavior [27]. Another aspect to consider in ASD sexuality concerns the different sensations and sensitivities compared to N-ASD sexuality, which can influence the modality of sexual physical contact [28].

Recently, it has been observed that there are significant differences between ASD and N-ASD individuals in various areas of sexual activity and sexual orientation. ASD individuals, especially women, are more likely to indicate sexuality with lower libido/sexual desire [29,30,31], higher asexuality rates [30,32], reduced heterosexuality [31,32,33], elevated hypersexual behavior/fantasies [34], and a higher incidence of “non-heterosexual” orientation (including homosexuality and bisexuality) [29,32,33].

A recent study with a very large sample found that individuals with high autistic traits tended to identify themselves more (1.73 times, 95% CI: 1.01–2.90) as bisexual or presented a sexuality not definable within the categories of heterosexual, homosexual, or bisexual [35]. Moreover, in a sample of women with ASD, there was found an increased prevalence of bisexual or homosexual orientation [36].

### Aims

This review aims to evaluate sexual function and behavior in individuals with ASD to foster a greater understanding of this important, although still often overlooked, issue.

## 2. Methods

### 2.1. Search Strategy

This review was conducted by searching recent peer-reviewed articles published between 1 June 2000 and 31 May 2022 using the following databases: PubMed, Embase, Cochrane Database, and Web of Science. A comprehensive search was conducted with search terms that included: (“Autism” OR “ASD”) AND (“Sexuality” OR “Romantic relation” OR “sexual behavior” OR “sexual awareness”).

### 2.2. Screening and Selection Process: Inclusion/Exclusion Criteria

After the removal of the duplicates, all articles were evaluated based on the titles and abstracts. The inclusion criteria were: (i) patients with ASD; (ii) sexual topic; (iii) English language; and (v) published in a peer-reviewed journal. We excluded articles that described theoretical models, methodological approaches, and basic technical descriptions. Furthermore, we excluded: (i) animal studies and (ii) conference proceedings or reviews. Papers dealing with sexual education and sexual victimization were also excluded. After an accurate revision of 741 full manuscripts, 32 articles satisfied the inclusion/exclusion criteria.

## 3. Results

### 3.1. Sexual Awareness

Sexual awareness is the knowledge and perception of feelings, desires, motivations, and situations related to sex. “Awareness” is essential to understand sexuality and sexual social relations, especially in individuals with developmental disorders [37]. Sexual awareness depends on the processes of cognitive attention and the ability to perceive sexual sensations and behaviors [38].

According to Snell et al. [38], awareness is based on four aspects: (1) sexual consciousness, which allows reflecting and thinking about your sexual properties since it consists in paying attention to one’s own signals of sexual arousal and motivation; (2) sexual monitoring that consists of the perception of the evaluation of others on one’s own sexuality; (3) sexual assertiveness, consisting in being assertive about one’s sexuality; and (4) consciousness of sexual appeal, which concerns the awareness of one’s own public sensuality.

These four areas in individuals with ASD are poorly developed, as they have low self-awareness and difficulty perceiving the mental states of others [39]. A study carried out by Hannah and Stagg found that subjects with ASD had significantly lower scores than N-ASD peers on all dimensions of sexual awareness, suggesting that young people with ASD should be educated about sexual aspects differently to peers [37]. Nonetheless, given that they may have difficulties initiating/interpreting sexual behavior and contextualizing sexual activities (e.g., masturbation), specific sexual coaching could help this patient population [40]. These results were recently confirmed by Pecora et al. The authors noted a growing awareness of the desire for sexual relationships in ASD individuals, although they pointed out that the relationship difficulties in ASD may cause a greater risk of engaging in inappropriate sexual behavior and sexual victimization than N-ASD peers [36]. Another interesting aspect of awareness is related to privacy rules [29,41,42]. In fact, Stokes and Kaur [41] found significant differences between ASD and N-ASD young people. ASD adolescents showed poorer social behaviors, fewer behaviors and knowledge about privacy rules, less sex education, and increased inappropriate sexual behavior. In addition, parents of ASD adolescents presented greater concerns than the N-ASD parents. Ginevra et al. have found that privacy awareness is lower in individuals with ASD than in N-ASD individuals [42]. As a result, patients affected by ASD may have less awareness of sexual situations and the privacy rules to be respected, with a greater risk of inappropriate or abusive behavior [42].

### 3.2. Sexual Identity and Gender Dysphoria

Gender is established on an anatomical basis at the time of birth according to the sexual genetic endowment, that is, on the basis of how the external genital organs present themselves. Gender does not always coincide with gender identity, which represents the perception that everyone has of themselves as male or female, or sometimes as belonging to categories other than male or female. In fact, gender dysphoria, i.e., affective and cognitive discomfort in relation to the gender that is anatomically assigned to us, can occur. It consists of a condition of separation between sex and gender identity [10], so it concerns the feeling of belonging to a different gender than the anatomical one, or the feeling of not entirely belonging to either the female or male gender or with fluid gender identity, oscillating over time between the feminine and the masculine. The concept of gender dysphoria was introduced in DSM-V to indicate the phenomenon of “gender inconsistency”. In particular, the criteria for identifying gender dysphoria are a marked inconsistency between experienced gender and primary/secondary sexual characteristics; a strong desire to get rid of one’s primary and/or secondary sexual characteristics; an intense desire for the sexual characteristics of the opposite gender, to belong to the opposite gender, and to be treated as a member of the opposite gender; the presence of a strong belief that you have feelings and reactions typical of the opposite gender; and, finally, the condition must be associated with clinically significant suffering [2].

In recent years, some authors have addressed the relationship between ASD and gender diversity and dysphoria [43,44,45,46,47,48,49,50,51,52,53]. Some researchers evaluated the co-occurrence of gender dysphoria and ASD [43,44,45,46,47,48,49,50,51,52,53], indicating that gender dysphoria and autism may be concomitant [50,52]. Other authors noted that the co-presence of gender dysphoria and ASD may be due to the unusual interests of ASD individuals, or as an expression of obsessive–compulsive disorder [49,51]. In any case, the results of these studies should be treated with caution, as many refer to case studies/series, and there are limitations in sampling and evaluation. Nevertheless, the current literature shows that ASD adolescents have a greater desire to belong to a gender other than their anatomical one compared to N-ASD adolescents [53]. Warrier et al. reported higher rates of autism diagnosis in transgender and gender-different adults than cisgender controls [54]. Estimates of gender dysphoria in N-ASD individuals range from 1:10,000 to 1:20,000 in men and 1:30,000 to 1:50,000 in women [55], whereas the prevalence rates in ASD individuals are in the range of 60 per 10,000 [56], and in some studies the prevalence is higher than 1% [57]. These results are in agreement with the literature, highlighting increased gender dysphoria in subjects diagnosed with autism [58,59]. Other studies have shown high levels of gender variance among ASD adults [60,61], as well as a high desire to belong to a gender other than the anatomical one [47]. Bejerot and Eriksson, in a study carried out on 50 adults with ASD and 53 controls, observed that men and women with ASD were classified with a less masculine gender role than controls. Furthermore, individuals with ASD reported atypical gender identity compared to controls [29]. In line with these results, a meta-analysis performed by Kallitsounaki and Williams highlighted a relationship between ASD traits and gender dysphoria in the general population [62].

### 3.3. Sexual Orientation

While gender identity is about the perception of oneself, sexual orientation consists of ways of relating to others and feeling romantic or sexual attraction for people of one gender rather than another. Sexual orientation does not coincide with gender: they are two different things, which can combine with each other in many ways and in different degrees [63]. Sexual orientation is determined by a mix of biological, environmental, and psychological factors. It persists over time, but sexual orientation and sexual identity are fluid in nature. In fact, one’s understanding or experience of one’s sexuality may change or evolve, and the orientation and sexual identity with it. Many people discover their sexual orientation at a young age, typically around puberty; for some, patterns of attraction, behavior, and self-identification remain stable throughout their lives, and for others these patterns develop more slowly.

Thus, sexual orientation is a complex construct that includes three main domains: attraction, contact, and identity, distinct from each other [64], and it is influenced by various socio-cultural factors [65]. There are myriad ways to describe sexual orientation, but the most common include: heterosexual, being attracted to the opposite gender; homosexual, being attracted to the same gender; and bisexual, being attracted to more than one gender. People who do not experience sexual attraction are sometimes called asexual; people who do not experience romantic attraction are sometimes called aromantic. However, because sexual orientation is complex and multifaceted, some find that a single term is inadequate to describe their experience and come up with their own new terms or combinations that they feel best describe them. Moreover, it is not correct to apply to “orientation” the statistical term of normality (that is what the majority do or profess); we prefer to talk in terms of prevalent sexual orientation (at least the one declared) and its physiological variants [66].

Some studies on ASD have shown higher rates of non-heterosexuality as compared to N-ASD peers [8,16,35,67,68]. Gilmour and coworkers observed that individuals with ASD have higher rates of asexuality, bisexuality, and homosexuality, as well as lower rates of heterosexuality than N-ASD controls [69]. An interesting finding was that females with ASD had less heterosexual orientation than males. This is also confirmed by another Swedish study in which females with ASD reported higher rates of homosexuality and bisexuality (58.3%) than controls (16%) [29]. Recently, George and Stokes carried out a study to evaluate sexual orientation between individuals with and without ASD [61]. The authors found that in all areas related to sexual orientation, males and females with ASD had fewer rates of heterosexuals, and reported more homosexuality, bisexuality, and asexuality than their sex-matched N-ASD peers. Moreover, in this study the authors report that “non-heterosexuality” is more pronounced among females than males with ASD. Another study carried out by the same authors confirmed these findings, highlighting that ASD individuals report an increase in homosexuality, bisexuality, and asexuality, but a decrease in heterosexuality [32].

In particular, the theme of asexuality is very interesting. Asexuality is a lack of sexual attraction for any gender. Some authors have found that many ASD individuals self-identify as asexual, possibly caused by deficits in social interaction and communication [30,70,71,72,73]. A recent review of the literature pointed out that asexuality and autism have similar aspects, such as the conception of the romantic dimensions, sexual attraction, and sexual orientation, as well as non-partner-oriented sexual desire [69]. Despite various findings, Ronis et al. suggest that researchers should be cautious about attributing higher asexuality rates among individuals with ASD than the general population by accurately assessing sexual identity [74].

Moreover, recent research has also found particular sexual orientations, such as emotional, romantic, and/or sexual attraction to inanimate objects (such as a bridge or a statue), termed objectophilia. Simner et al. found in 34 individuals with objectophilia that sexual orientation could be linked to autism and synaesthesia [75].

### 3.4. Sexual Behavior

Subjects with ASD may experience difficulties in multiple social areas, such as physical contact, self-understanding, and social interaction, which lead to stereotyped or ritualized mannerisms. However, various studies have shown that many ASD adults may exhibit healthy sexual functioning and experience sexual intimacy [8,16]. Despite these studies, the sexual behavior of people with ASD is often overlooked. Usually, sexual feelings are expressed through masturbation, alone or in the presence of other people. Instead, sexual activity toward a person is represented by touching or kissing the “partner”, often in an unwanted way [24]. A study carried out by Hellemans et al. indicated that individuals with high-functioning ASD may express both sexual interest and various sexual behaviors [24]. In fact, the authors identified a high interest in relationships and the establishment of relationships with age-appropriate sexual behavior. In particular, masturbation occurred in 80% of male adolescents in the 14–15-year-old age group and in 90% in the 16–19-year-old age group, as confirmed by previous studies [76]. Furthermore, Byers et al. in a survey on individuals with ASD (age 21–73, mean age 35.3) found that 59% of the group had a romantic relationship of at least 3 months [16]. Dewinter et al. observed that a couple of sexual acts in ASD males had the same frequency as those in non-ASD subjects [26]. These results were confirmed also by Simcoe et al. [77]. However, Hartmann found that ASD young adults seek more privacy and engage in more typical sexual behaviors and higher sexual victimization than their parents reported [78].

Another aspect highlighted in some studies is that ASD individuals tend to exhibit more deviant sexual behavior than their N-ASD peers. In these cases, it is important to separate typical sexual behaviors, such as general sexual curiosity, masturbation, and interest in the genitals of peers or siblings, from uncommon behaviors, such as explicit imitation of sexual acts, asking to engage in sexual activity, inserting objects into the genitals, and frequent sexual behaviors that are resistant to distraction [79,80].

In particular, excessive masturbation [81,82]; exhibitionist behaviors [83]; pedophilic fantasies or behaviors [84]; fetishistic fantasies or behaviors [85]; sadomasochism [86] or other forms of paraphilias [87]; and hypersexuality [34] have been reported. Thus, it is clear that ASD subjects have adequate sexual development and sexual behavior, even if in some cases it is more deviant than that of N-ASD subjects.

#### Risky Sexual Behavior and Sexually Transmitted Infection

Some authors highlighted that although individuals with disabilities, compared to their peers, may have fewer and delayed sexual relationships, they are more likely to engage in risky sexual behaviors [34,88,89]. Since sexually active adolescents may be at increased risk for sexually transmitted diseases, it would be helpful to promote the use of protective barriers, such as reversible and appropriate long-acting contraception [90,91,92]. Unfortunately, this topic in the literature relating to adolescents with ASD has been poorly considered. Li et al. conducted a longitudinal study between 2001 and 2009 of 5076 adolescents and young adults with ASD and 57,060 individuals of the same age/sex without ASD. The results showed that patients with ASD tended to manifest a higher prevalence of sexually transmitted infections (hazard ratio (HR) 3.36) than the control group [93]. Various opportunities may exist to promote individualized sexual safety and education based on sexual orientation and to prevent problematic sexual behavior. From this point of view, it has been shown that typical sex education may not be sufficient for people with ASD, but specific methods are needed to meet needs in personalized modalities [37,94].

In Table 1, individuals with ASD present with reduced sexual awareness, a high prevalence of gender dysphoria, and alterations in sexual behavior.

## 4. Discussion

Individuals with ASD may have the same sexual desires as N-ASD peers. However, it has been shown that, concerning sexuality and romantic relationships, individuals with ASD face unique challenges in relating to peers and other individuals [40,96]. Our review revealed that adolescents with ASD may experience higher prevalence rates of inappropriate sexual behaviors and gender dysphoria. Furthermore, sexual awareness is reduced in this patient population and the prevalence of other physiological variants of sexual orientation (e.g., homosexuality, asexuality, bisexuality, etc.) is higher in adolescents with ASD than in N-ASD peers.

An interesting aspect of the literature is that the findings regarding sexual orientation (attraction to someone of the same sex, the other sex, or both sexes) in adolescents and adults with ASD are varied. Some studies found a higher prevalence of homosexuality (5–10%) among ASD participants [26,68], while other studies reported higher levels of asexuality in ASD [69,74] compared to the general population. This discrepancy may be due to the different tools used to investigate sexuality as well as to other biases in the methodology of the investigation. A recent study on 67 ASD individuals who identify with sexual minorities found that autistic traits complicate self-identification of sexual orientation. In addition, ASD subjects’ sexual manifestations may be affected by difficulty recognizing and communicating sexual needs, and then they tend to feel misunderstood and isolated [95]. The authors underlined that ASD individuals can experience a “double minority”, which is a difficult issue to manage for them because it affects the formation of identity, with greater vulnerability in sexual relationships [95]. Indeed, adolescents and adults with ASD and LGBT orientation (i.e., lesbian, gay, bisexual, transgender) can experience a sense of difference in their ASD characteristics, gender identity, and sexual orientation [97]. It is important to increase knowledge about sexual orientation and identity in adolescents and adults with ASD to foster awareness and personalization in sexual education and support.

Another aspect that emerged from our review is the relationship between ASD and gender dysphoria [32,43,47]. Gender, as mentioned, consists of behaviors and attitudes considered “typical” for males/females in a specific culture and historical period [98]. It has been shown that ASD traits in people with gender dysphoria are relatively high [53,59]. Different motivations have been put forward to explain the higher prevalence of variations in sexual orientation and gender identity in people with ASD [32,43,47]. For example, biological theories claim that dysphoria and ASD exhibit common genetic patterns, whilst psychosocial theories believe they are due to social experiences, opportunities to meet people of the opposite sex, sensory preferences, stereotyped interests, and limited flexibility. However, the results for ASD and gender dysphoria were conducted on samples that exhibited different characteristics between subjects with ASD and N-ASD, so attention should be paid to this when considering the studies.

Regarding sexual expression, recent studies have highlighted that sexual experiences, both alone and in pairs, are common for most adolescents and adults with high-functioning ASD [2,9,16,26,69,74], in contrast to previous research in which subjects with ASD were thought to be devoid of sexuality or with problematic sexuality [8,99,100,101]. For example, Hellemans et al. found that 96% of the ASD sample expressed an interest in sexuality [24], while Gotham et al. found that 47% reported wanting a romantic partner [102]. Several studies have shown that adults with ASD are less likely to be in a relationship [30,31,33,67] than their N-ASD peers. Due to the main symptoms of ASD (such as social and interpersonal skills deficit) and the lack of adequate sex education, various authors have shown that some individuals with ASD may develop non-normative and problematic sexual behaviors, such as hypersexuality or paraphilias [34,40,44]. Indeed, it should be noted that certain behavioral and information-processing characteristics in individuals with ASD can directly influence the sexual experience. Furthermore, as already stated, limited sexual knowledge and experience, combined with social deficits, can expose individuals with ASD to a high risk of victimization [21]. Brown-Lavoie et al. investigated 95 adults with ASD and 117 adults without ASD for sexual knowledge, actual experiences, and sexual victimization. It was found that individuals with ASD had less sexual knowledge, drawn from non-social sources (e.g., via internet resources), had less effective knowledge, and reported experiencing greater sexual victimization than controls [21]. This aspect, despite being outside the scope of the review and therefore not investigated, is an important issue that deserves future attention.

## 5. Limitations of the Studies and Future Research

Studies in the literature have several methodological limitations. First, most of the existing research is based on relatively small samples (the largest recent studies included *n* = 141 [16] and *n* = 229 [73]), and on case studies. Moreover, only more recent studies are based on self-evaluation, while most of the previous ones include surveys with interviews with parents on the sexuality of their children. These aspects have reduced the validity of the studies and their generalizability. Finally, the recruitment of volunteers for research on sexuality may also have skewed the results. It has been shown that volunteers in this research area usually have more sexual experience and value more positively sexuality [72]. Further research with more rigorous methods and more homogeneous samples would be useful to increase the knowledge of sexuality in ASD individuals and to favor the correct management of this aspect in their lives.

To this end, all adolescents with ASD should receive adequate sexual education, which should involve not only the patients but also their parents/families and healthcare professionals to overcome social barriers and prejudice. Proper education should promote sexual health and favor peer relationships, avoiding isolation and helping in the management of sexual dynamics in order to ensure the well-being and quality of life of ASD people [36]. In this context, the current studies on robots applied to the education of subjects with autism are interesting and futuristic. It was pointed out that robots could offer various opportunities for people with ASD [103,104,105,106]. Studies have shown that simple robots with human characteristics can facilitate interaction with ASD subjects more than human trainees [104,106]. In fact, robots seem to be more accepted for the ease of engaging in interactions. Hence, this medium could also be used to help define proper sex education. Pennisi et al. showed positive effects in the use of social robots in ASD individual therapy: the subjects with ASD tended to have better interactions with a robot partner; moreover, during robotic sessions, the ASD subjects showed a reduction in repetitive and stereotyped behavior and better spontaneous language [105]. Therefore, robots can also be an adequate means to interact with and provide knowledge more easily to autistic subjects [107]. Nonetheless, currently only a few studies have been conducted in the field of robotics on ASD subjects, and their application in sex education/coaching raises various technical, clinical, and ethical concerns that should be investigated to provide these vulnerable individuals with the best training possible.

## 6. Conclusions

In conclusion, this systematic review shows that individuals with ASD present a high prevalence of sexual orientation variants, inappropriate behaviors, and gender dysphoria as compared to peers. Although these individuals have multiple sexual interests and desires for intimate relationships, it is evident that a satisfactory and healthy expression of their sexuality is affected by various factors. In fact, ASD sexuality and identity can be influenced by deficits in communication and social interaction, as well as by the limitations present in the social environment itself, which does not allow adequate social experiences and contacts compared to N-ASD peers. In this way, both social obstacles and poor sex education do not allow young people and adults with ASD to live their sexuality smoothly, regardless of their sexual orientation [104]. Thus, the need for an adequate, personalized, and contextualized sexual education tailored to the characteristics and needs of ASD subjects is fundamental.

## Figures and Tables

**Table 1 brainsci-12-01427-t001:** Shows the main studies concerning sexuality in autism spectrum disorders.

Study	Study’s Design	Patients	Major Findings
Weir et al. [1]	Observational study	2386 adults (*n* = 1183 ASD)	The study pointed out that autistic adults are interested in sexual relationships and sexual activity.
Dewinter et al. [9]	Observational study	30 ASD adolescents 60 individuals	The authors found that most of the teens in the control and ASD groups reported masturbating and experiencing an orgasm. Adolescents with ASD reported fewer sexual experiences in pairs than in the control group.
Byers et al. [16]	Observational study	141 ASD adults	These results highlight the importance of research on and sexuality education for individuals with ASD to enhance sexual well-being.
Holmes et al. [17]	Observational study	190 parents of ASD adolescents	The authors emphasize the importance of parents as a primary source of sex education for adolescents with ASD. They highlighted that sex education must be adapted to the child’s developmental level.
Sala et al. [18]	Observational study	31 ASD subjects 26 N-ASD individuals	The study found that ASD and N-ASD individuals have similar notions of intimacy, but ASD subjects experience greater uncertainty about relationships and communication than N-ASD peers.
Ousley et al. [20]	Observational study	21 high-functioning ASD adults 20 mildly to moderately mentally retarded adults without autism	The study found group differences in the sexual experience, greater among adults with mental retardation, but not in knowledge or interest.
Brown-Lavoie et al. [21]	Observational study	95 adults with ASD 117 adults without ASD	The authors noted that individuals with ASD have greater sexual knowledge from nonsocial sources and are more at risk of sexual victimization than controls.
Stokes et al. [23]	Observational study	25 ASD adolescents and adults 38 typical adolescents and adults	The ASD group relied less on peers and friends for social and romantic learning. Individuals with ASD were more likely to engage in inappropriate courtship behavior and were more likely to focus their attention on celebrities, strangers, co-workers, and former partners than controls.
Hellermans et al. [24]	Observational study	24 caregivers of ASD individuals	The authors found that masturbation was common. The number of bisexual orientations appeared high. Ritual–sexual use of objects and sensory fascination were present. Paraphilia was present in 2 subjects.
Dewinter et al. [26]	Observational study	50 adolescent boys with ASD	Results demonstrated substantial similarity between the groups in terms of sexual behaviors. The only significant difference was that boys with ASD were more tolerant towards homosexuality compared to the control group.
Bejerot and Eriksson [29]	Case–control study	50 adults with ASD 53 N-ASD subjects	The authors observed that men and women with ASD were classified with a less masculine gender role than controls. Furthermore, individuals with ASD reported atypical gender identity compared to controls.
Bush et al. [30]	Brief report	247 ASD individuals	Asexual participants indicated less sexual desire and behaviors than subjects with other sexual orientations, but greater sexual satisfaction.
Pecora et al. [31]	Observational study	231 ASD individuals 161 N-ASD controls	ASD females reported less sexual interest but more experiences than autistic males. ASD women are more at risk for negative sexual experiences including victimization and abuse than ASD men.
George and Stokes [32]	Observational study	309 ASD individuals 310 N-ASD individuals	The ASD group had more homosexuality, bisexuality, and asexuality, and less heterosexuality.
Dewinter et al. [33]	Qualitative study	8 ASD adolescents	The authors pointed out that adolescent ASD individuals experience their sexuality.
Hannah and Stagg [37]	Observational study	20 N-ASD 20 ASD individuals	The authors underlined that subjects with ASD had significantly lower scores than N-ASD peers on all dimensions of sexual awareness.
Stokes and Kaur [41]	Observational study	50 parents of N-ASD individuals 23 parents of ASD adolescents	The parents of the two groups differed significantly in all dimensions considered. ASD parents should be supported with specific educational programs.
Ginevra et al. [42]	Observational study	94 parents of N-ASD adolescents 93 parents of adolescents with Down’s syndrome 82 parents of ASD adolescents	The authors demonstrated that ASD individuals may have less awareness of sexual situations and the privacy rules to be respected, with a greater risk of inappropriate or abusive behavior.
Hendriks et al. [33]	Observational study	89 ASD adults	The results are consistent with the literature highlighting increased gender dysphoria in subjects diagnosed with autism.
van Der Miesen et al. [47]	Observational study	573 adolescents and 807 adults with ASD 1016 adolescent and 846 adult N-ASD subjects	No significant gender differences were found in adults with ASD.
de Vries et al. [53]	Observational study	204 individuals with gender identity disorder	The incidence of ASD in this sample of children and adolescents was 7.8% (*n* = 16).
Warrier et al. [54]	Observational study	5 cross-sectional datasets consisting of 641,860 individuals	The authors found that transgender and gender-diverse individuals scored higher on self-assessment measures of autistic traits and lower on empathy measures of self-assessment.
Jones et al. [58]	Observational study	61 transsexuals 198 transsexual women 76 typical males 98 typical females 125 individuals with Asperger’s syndrome	Trans men have more autistic traits and may have had difficulty socializing with peers and therefore have found it easier to identify with male peer groups.
Cooper et al. [60]	Observational study	219 ASD individuals 267 controls	The authors showed that autistic people, particularly females, had less social identification with a gender group and more negative feelings.
George and Stokes [61]	Observational study	309 ASD subjects 261 N-ASD individuals	The authors pointed out that there is a deterioration in mental health among individuals with ASD, especially if they belong to sexual and gender minorities compared to heteronormative populations.
Fernandes et al. [68]	Observational study	184 ASD adolescents and young adults	The authors found that the majority of the sample had a sexual interest and interest in the opposite sex. Inappropriate sexual behavior and paraphilias were reported for about a quarter of individuals.
Gilmour, Schalom, Smith [69]	Observational study	82 (55 female and 17 male) adults with ASD	The results suggested a higher degree of homosexuality among females with ASD, although this effect did not reach significance.
Strunz et al. [73]	Observational study	29 high-functioning adults with ASD	The study findings showed that the majority of ASD high-functioning adults are interested in romantic relationships.
Ronis et al. [74]	Observational study	332 ASD individuls	17 (5.1%) participants who met the study criteria (*n* = 332) self-identified as asexual.
Simner et al. [75]	Observational study	34 objectum-sexuality individuals 88 controls	Objectum-sexuality individuals possess high rates of autistic traits compared to controls and a high prevalence of significant synaesthetic traits inherent in the nature of their attractions.
Simcoe et al. [76]	Observational study	111 parents of ASD children 212 parents of N-ASD children	The results highlighted that gender behavior, sensory sensitivity, conforming behaviors, imagination, and imitation subscales differentiated autistic females from N-ASD females.
Lewis et al. [95]	Observational study	67 individuals who identified as autistic sexual minorities	Autistic sexual minorities can experience a “double minority”, which affects identity formation and increases vulnerability in sexual relationships.

Autism spectrum disorder (ASD); non-autism spectrum disorder (N-ASD).

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
