# Peer review of "Sex and Sexuality in Autism Spectrum Disorders: A Scoping Review on a Neglected but Fundamental Issue"

_brainsci, 2022, doi:10.3390/brainsci12111427_

Round 1
Reviewer 1 Report
This paper presents a small review on a limited number of papers concerning autism and sexuality published between 2000 and 2022. It concludes that there is a shortage of research in the area, and that while autistic individuals desire sexual relations in a manner similar to non-autistic individuals, the heightened prevalence of non-heterosexual orientations and diverse genders renders their sexuality abnormal.
I cannot recommend that this paper be further process in the current format with the current uses of language be published. At the very least, the paper must adjust its language and structure its definitions before it concludes things to be abnormal or normal.
For instance, in the abstract, it is stated that “their sexuality is often abnormal because of higher prevalence rates of gender dysphoria and inappropriate sexual behavior.” How sexuality is rendered abnormal by an individual having either diverse gender orientation, or by them having a non-heterosexual orientation? This assertion is at least inconsiderate, and quite possibly offensive. There is no definition of normality offered that is violated by these orientations, there is no reason to conclude that a diverse orientation in either gender or sexuality is abnormal. Further, normal sexuality in the current period includes diverse sexuality and diverse genders.
In the opening paragraph, autism is presented as affecting 1:100 individuals, with a ratio of 4 males to each female diagnosed. Few authors would currently hold with such outdated estimates of prevalence or the gender ratio. The citation associated with these outdated claims is the DSM-5, which was published in 2013 and was itself then considered at best controversial, but more widely accepted as too conservative. More recent estimates can easily be found that place estimates of prevalence close to 2% (see https://www.cdc.gov/ncbddd/autism/index.html), and with gender ratios closer to 2.5:1.
The description of autism as a disease ignores the last 30 years of conceptualization around autism. It has not been seen as a “disease” for at least this period. It is a developmental condition. This paper’s characterization as “disease” is offensive, and should be restated. Moreover, on page 3, non-autistic people are described as “able-bodied”. Autism is not a condition generally regarded to afflict the body. So in what sense are autistic people “disabled bodies”? The correct term for non-autistic people is “non-autistic”. Also, this term is used in relation to work presented by Ginerva et al. who do not use the term “able-bodied”. Last, the work attributed to Ginerva et al. was in fact a replication of work published much earlier by Stokes and Kaur (one of the many missing papers in this review https://doi.org/10.1177/1362361305053258).
The blanket description of autism offered on page 1 as being “and young patients may be distant, numb, and with poor ability to interact with others [2].” This description was only applied to a proportion of those with autism by Prior and Ozonoff, and did not intend that this was a blanket description of all autistic persons.
The attribution to Dewinter that “However, sexuality in adolescents and young with ASD is still a poorly investigated and neglected issue” In 2013, when Dewinter wrote those words, there were about 240 published papers on this topic. Now there are about 1000. I do not think it is a reasonable claim to make that the area is under-researched. Further, a casual glance through the reference list of this paper reveals it is missing many central and crucial papers. In using the same search strategy as applied by the authors, I located many papers not included that met all inclusion criteria as specified.
On page 4, it is asserted that “The few studies on ASD have shown higher rates of non-heterosexuality as compared to able-bodied peers [4,12,44,45]”. There have been substantially more studies undertaken of diverse sexuality. Further, sexuality should not be defined as being heterosexual and non-heterosexual, and autism should not be defined as not-able bodied. Please use appropriate descriptors throughout.
The claim on p4 that Dewinter's work was based upon interviews with the individuals, and were not proxy interviews is false: “These data appear to be in contrast to the previous literature, but this is probably due to the different methodology used, being interviews with parents on the sexuality of ASD children the most used”. Dewinter interviewed his participants.
On page 6 it is asserted that “although individuals with ASD may have a “normal” sexual functioning, their sexuality is often abnormal because of higher prevalence rates of gender dyshoria and inappropriate sexual behavior”. I disagree that this review has demonstrated this at all.
On page 6 it is asserted that “The sexual development of individuals with ASD differs from that of typically developing peers [58], probably because some characteristics of ASD in adolescents affect sexual functioning in different ways.” This assertion is flawed and is based upon very out of date data and an old and uninformed perspective. The cited work [58] was a chapter published in 2007, and probably written in 2003 to 2004 before almost all of the modern understanding of autism was obtained, and before all the modern understanding about sexuality in autism was obtained.
Page 6 “First, the characteristics of the groups studied differ in the intellectual abilities of the participants, and in the characteristics of autism.” Of most of the studies cited, and most of those not cited, rely upon what used to be termed high functioning autism as participants. Their intellectual levels are not different from the non-autistic population. Hence, to make this claim is simply wrong. The authors would need to stipulate the exact evidence to which they refer.
Page 6 “Finally, the tests used were mostly based on parental interviews regarding the sexuality of their children”. This is only true of three of the 11 papers covered in the table, and is not widely true of the literature, where I can easily locate in excess of 40 relevant papers suitable for inclusion in this study, among which few rely upon proxy samples.
Page 6 it is also stated that “Another important issue concerns the poor knowledge of sexuality by healthcare professionals, as sexual function in patients with ASD as well as other neuro-developmental disabilities is often disregarded, though it is a topic of great importance to patients and to those with whom they share significant relationships”. It is not clear to me how the ignorance of some health care providers damages the credibility of research undertaken by researchers who are well aware of the issues.
Author Response
This paper presents a small review on a limited number of papers concerning autism and sexuality published between 2000 and 2022. It concludes that there is a shortage of research in the area, and that while autistic individuals desire sexual relations in a manner similar to non-autistic individuals, the heightened prevalence of non-heterosexual orientations and diverse genders renders their sexuality abnormal. I cannot recommend that this paper be further process in the current format with the current uses of language be published. At the very least, the paper must adjust its language and structure its definitions before it concludes things to be abnormal or normal.
Thank you for your opinion, we reviewed the manuscript, as suggested. Indeed, we have added much more literature (and the missing studies in the table), and completely revised definitions, avoiding to let the readers misunderstand “abnormality” in sexual esxpression.
For instance, in the abstract, it is stated that “their sexuality is often abnormal because of higher prevalence rates of gender dysphoria and inappropriate sexual behavior.” How sexuality is rendered abnormal by an individual having either a diverse gender orientation or by them having a non-heterosexual orientation? This assertion is at least inconsiderate, and quite possibly offensive. There is no definition of normality offered that is violated by these orientations, there is no reason to conclude that a diverse orientation in either gender or sexuality is abnormal. Further, normal sexuality in the current period includes diverse sexuality and diverse genders.
We reviewed the text, as suggested. Indeed, we want to clarify that abnormal was referred only to behaviors and not to orientation, where now we have specified that homesexuality, bisexualiry,... are physiological variants of orientation.
In the opening paragraph, autism is presented as affecting 1:100 individuals, with a ratio of 4 males to each female diagnosed. Few authors would currently hold with such outdated estimates of prevalence or the gender ratio. The citation associated with these outdated claims is the DSM-5, which was published in 2013 and was itself then considered at best controversial, but more widely accepted as too conservative. More recent estimates can easily be found that place estimates of prevalence close to 2% (see https://www.cdc.gov/ncbddd/autism/index.html), and with gender ratios closer to 2.5:1.
We corrected the old data on prevalence and added those to whom the referee correctly suggested.
The description of autism as a disease ignores the last 30 years of conceptualization around autism. It has not been seen as a “disease” for at least this period. It is a developmental condition. This paper’s characterization as “disease” is offensive, and should be restated.
It was out of our intention to offend people with ASD by using the “old” terminology. We have now corrected and rebuilt all the periods in which this mistake was present, and we have used concepts and terminology from the last researches.
Moreover, on page 3, non-autistic people are described as “able-bodied”. Autism is not a condition generally regarded to afflict the body. So in what sense are autistic people “disabled bodies”? The correct term for non-autistic people is “non-autistic”. Also, this term is used in relation to work presented by Ginerva et al. who do not use the term “able-bodied”. Last, the work attributed to Ginerva et al. was in fact a replication of work published much earlier by Stokes and Kaur (one of the many missing papers in this review High-functioning autism and sexuality: A parental perspective - Mark A. Stokes, Archana Kaur, 2005).
We reviewed the language used in the manuscript, avoiding the old terminology and adding N-ASD for non autistic peers.
The blanket description of autism is offered on page 1 as being “and young patients may be distant, numb, and with poor ability to interact with others [2].” This description was only applied to a proportion of those with autism by Prior and Ozonoff, and did not intend that this was a blanket description of all autistic persons.
We removed this sentence, and the diverse degree of the syndrome specified.
The attribution to Dewinter that “However, sexuality in adolescents and young with ASD is still a poorly investigated and neglected issue” In 2013, when Dewinter wrote those words, there were about 240 published papers on this topic. Now there are about 1000. I do not think it is a reasonable claim to make that the area is under-researched. Further, a casual glance through the reference list of this paper reveals it is missing many central and crucial papers. In using the same search strategy as applied by the authors, I located many papers not included that met all inclusion criteria as specified.
We revised the text and added the missing references, by re-revising the literature on the field taking into consideration this important concern by the reviewer.
On page 4, it is asserted that “The few studies on ASD have shown higher rates of non-heterosexuality as compared to able-bodied peers [4,12,44,45]”. There have been substantially more studies undertaken of diverse sexuality. Further, sexuality should not be defined as being heterosexual and non-heterosexual, and autism should not be defined as not-able bodied. Please use appropriate descriptors throughout.
We have revisited the text as suggested by using the proper terminology.
The claim on p4 that Dewinter's work was based upon interviews with the individuals, and were not proxy interviews is false: “These data appear to be in contrast to the previous literature, but this is probably due to the different methodology used, being interviews with parents on the sexuality of ASD children the most used”. Dewinter interviewed his participants.
We corrected this mistake.
On page 6 it is asserted that “although individuals with ASD may have a “normal” sexual functioning, their sexuality is often abnormal because of higher prevalence rates of gender dyshoria and inappropriate sexual behavior”. I disagree that this review has demonstrated this at all.
We corrected this sentence and specified in the discussion what literature found on this issue.
On page 6 it is asserted that “The sexual development of individuals with ASD differs from that of typically developing peers [58], probably because some characteristics of ASD in adolescents affect sexual functioning in different ways.” This assertion is flawed and is based upon very out of date data and an old and uninformed perspective. The cited work [58] was a chapter published in 2007, and probably written in 2003 to 2004 before almost all of the modern understanding of autism was obtained, and before all the modern understanding about sexuality in autism was obtained.
We corrected this sentence, and better specified the difference in sexual expressions and behavior as per new literature data.
Page 6 “First, the characteristics of the groups studied differ in the intellectual abilities of the participants, and in the characteristics of autism.” Of most of the studies cited, and most of those not cited, rely upon what used to be termed high functioning autism as participants. Their intellectual levels are not different from the non-autistic population. Hence, to make this claim is simply wrong. The authors would need to stipulate the exact evidence to which they refer.
We have rephrased the sentence, and made it more clear. However, both introduction and discussion have been totally rebuilt to avoid misunderstanding concepts and terminology.
Page 6 “Finally, the tests used were mostly based on parental interviews regarding the sexuality of their children”. This is only true of three of the 11 papers covered in the table, and is not widely true of the literature, where I can easily locate in excess of 40 relevant papers suitable for inclusion in this study, among which few rely upon proxy samples.
We rewrote this sentence, and specified the tools used..
Page 6 it is also stated that “Another important issue concerns the poor knowledge of sexuality by healthcare professionals, as sexual function in patients with ASD as well as other neuro-developmental disabilities is often disregarded, though it is a topic of great importance to patients and to those with whom they share significant relationships”. It is not clear to me how the ignorance of some health care providers damages the credibility of research undertaken by researchers who are well aware of the issues.
We removed these sentences, and specified the need for more knowledge about this issue, nor only for ASD individuals but also their caregivers and healthcare professionals .
Reviewer 2 Report
This is a very interesting paper focusing on sex and sexuality in patients with autism spectrum disorders. the paper is well-written and of interest for the journal. However, several minor changes should be made before considering it for publication.
Abstract:
1- The search terms used in the present review should be also described in the abstract section. Other details about the methods have been well-described (databases, etc).
Introduction
1-The introduction is really brief. I recommend to expand it by focusing on social cognition, interpersonal relationships and then, by focusing in more detail with sexual behaviors.
2- The main aims of the present review merits a separate subsection in the introduction. 1.1. "Aims".
Methods
1- Search strategy should be a subsection of the methods section, not a separate section.
2- I recommend to add some subsections, for instance: screening and selection process, databases (sources), inclusion/exclusion criteria.
Results
1- Use of contraceptives and other preventive strategies should be considered also in the present review. Please, add a brief paragraph or subsection after the "Sexual Behavior" subsection.
Discussion
1- "The authors point of view" should be renamed as Discussion.
2- A conclusions section is necessary, as well a limitations and strentghs section. Please, separate the discussion from the conclusions.
Conclusions:
1-A brief conclusion on sexual identity, sexual orientation and behavior should be described.
2- What about future directions for investigation? What are the authors recommending to clinicians with respect to the results from this review?
Author Response
This is a very interesting paper focusing on sex and sexuality in patients with autism spectrum disorders. the paper is well-written and of interest for the journal. However, several minor changes should be made before considering it for publication.
Thank you for your opinion, we reviewed the manuscript as suggested.
Abstract:
1- The search terms used in the present review should be also described in the abstract section. Other details about the methods have been well-described (databases, etc).
Done.
Introduction
1-The introduction is really brief. I recommend to expand it by focusing on social cognition, interpersonal relationships and then, by focusing in more detail with sexual behaviors.
We enlarged introduction, as suggested.
2- The main aims of the present review merits a separate subsection in the introduction. 1.1. "Aims".
Done.
Methods
1- Search strategy should be a subsection of the methods section, not a separate section.
Done.
2- I recommend to add some subsections, for instance: screening and selection process, databases (sources), inclusion/exclusion criteria.
Done.
Results
1- Use of contraceptives and other preventive strategies should be considered also in the present review. Please, add a brief paragraph or subsection after the "Sexual Behavior" subsection.
Done, as suggested.
Discussion
1- "The authors point of view" should be renamed as Discussion.
Done.
2- A conclusions section is necessary, as well limitations and strengths section. Please, separate the discussion from the conclusions.
Done.
Conclusions:
1-A brief conclusion on sexual identity, sexual orientation and behavior should be described.
2- What about future directions for investigation? What are the authors recommending to clinicians with respect to the results from this review?
We redefined the last paragraph: discussion limits; future directions and conclusions.